# NEEDLES IN HAYSTACKS:
# ON CLASSIFYING TINY OBJECTS IN LARGE IMAGES

## ABSTRACT

In some important computer vision domains, such as medical or hyperspectral imaging, we care about the classification of *tiny objects* in large images. However, most Convolutional Neural Networks (CNNs) for image classification were developed using biased datasets that contain *large objects*, in mostly central image positions. To assess whether classical CNN architectures work well for tiny object classification we build a comprehensive testbed containing two datasets: one derived from MNIST digits and one from histopathology images. This testbed allows controlled experiments to stress-test CNN architectures with a broad spectrum of signal-to-noise ratios. Our observations indicate that: (1) There exists a limit to signal-to-noise below which CNNs fail to generalize and that this limit is affected by dataset size — more data leading to better performances; however, the amount of training data required for the model to generalize scales rapidly with the inverse of the object-to-image ratio (2) in general, higher capacity models exhibit better generalization; (3) when knowing the approximate object sizes, adapting receptive field is beneficial; and (4) for very small signal-to-noise ratio the choice of global pooling operation affects optimization, whereas for relatively large signal-to-noise values, all tested global pooling operations exhibit similar performance.

## 1 INTRODUCTION

Convolutional Neural Networks (CNNs) are the current state-of-the-art approach for image classification (Krizhevsky et al., 2012; Simonyan & Zisserman, 2014; He et al., 2015; Huang et al., 2017). The goal of image classification is to assign an image-level label to an image. Typically, it is assumed that an object (or concept) that correlates with the label is clearly visible and occupies a *significant* portion of the image (Lecun et al., 1998; Krizhevsky, 2009; Deng et al., 2009). Yet, in a variety of real-life applications, such as medical image or hyperspectral image analysis, only a small portion of the input correlates with the label, resulting in low signal-to-noise ratio. We define this input image signal-to-noise ratio as Object to Image (O2I) ratio. The O2I ratio range for three real-life datasets is depicted in Figure 1. As can be seen, there exists a distribution shift between standard classification benchmarks and domain specific datasets. For instance, in the ImageNet dataset (Deng et al., 2009) objects fill at least $1\%$ of the entire image, while in histopathology slices (Ehteshami Bejnordi et al., 2017) cancer cells can occupy as little as $10^{-6}\%$ of the whole image.

Recent works have studied CNNs under different noise scenarios, either by performing random input-to-label experiments (Zhang et al., 2017; Arpit et al., 2017) or by directly working with noisy annotations (Mahajan et al., 2018; Jiang et al., 2017; Han et al., 2018). While, it has been shown that large amounts of label-corruption noise hinders the CNNs generalization (Zhang et al., 2017; Arpit et al., 2017), it has been further demonstrated that CNNs can mitigate this label-corruption noise by increasing the size of training data (Mahajan et al., 2018), tuning the optimizer hyperparameters (Jastrzębski et al., 2017) or weighting input training samples (Jiang et al., 2017; Han et al., 2018). However, all these works focus on input-to-label corruption and do not consider the case of noiseless input-to-label assignments with low and very low O2I ratios.

In this paper, we build a novel testbed allowing us to specifically study the performance of CNNs when applied to tiny object *classification* and to investigate the interplay between input signal-to-noise ratio and model generalization. We create two synthetic datasets inspired by the children's puzzle book *Where's Wally?* (Handford, 1987). The first dataset is derived from MNIST digits and allows us

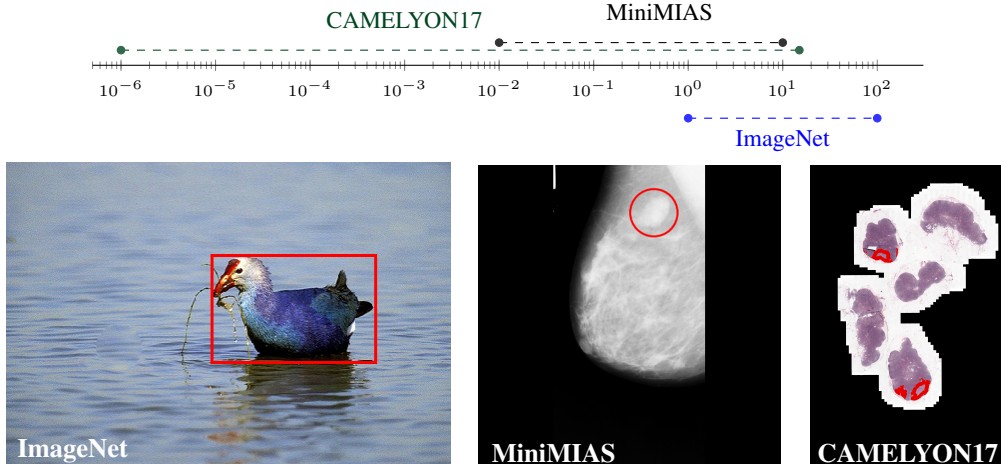

Figure 1: Range of Object to Image (O2I) ratios [%] for two medical imaging datasets (CAME-LYON17 (Ehteshami Bejnordi et al., 2017) and MiniMIAS (Suckling, 1994)) as well as one standard computer vision classification dataset (ImageNet (Deng et al., 2009)). The ratio is defined as $O2I = \frac{A_{object}}{A_{image}}$, where $A_{object}$ and $A_{image}$ denote the area of the object and the image, respectively. Together with O2I range, we display examples of images jointly with the object area $A_{object}$ (in red).

to produce a relatively large number of datapoints with explicit control of the O2I ratio. The second dataset is extracted from histopathology imaging (Ehteshami Bejnordi et al., 2017) where we crop images around lesions and obtain small number of datapoints with an approximate control of the O2I ratio. To the best of our knowledge these datasets are the first ones designed to explicitly stress-test the behaviour of the CNNs in the low input image signal-to-noise ratio.

We develop a classification framework, based on CNNs, and analyze the effects of different factors affecting the model optimization and generalization. Throughout an empirical evaluation, we make the following observations:

- Models can be *trained in low O2I regime without using any pixel-level annotations and generalize* if we leverage enough training data. However, *the amount of training data required for the model to generalize scales rapidly with the inverse of the O2I ratio*. When considering datasets with fixed size, we observe an *O2I ratio limit* in which all tested scenarios fail to exceed random performance.
- We empirically observe that *higher capacity models show better generalization.* We hypothesize that high capacity models learn the input noise structure and, as result, achieve satisfactory generalization.
- We confirm the importance of model inductive bias — in particular, the *model's receptive field size*. Our results suggest that different pooling operations exhibit similar performance, for larger O2I ratios; however, for very small O2I ratios, the type of *pooling operation affects the optimization ease*, with max-pooling leading to fastest convergence.

The code of our testbed will be publicly available at: `https://anonymous.url` allowing to reproduce all data and results; we hope this work can serve as a valuable resource facilitating further research into the understudied problem of low signal-to-noise classification scenarios.

## 2 TESTBED FOR LOW SIGNAL-TO-NOISE CLASSIFICATION SCENARIOS

### 2.1 DATASETS: IS THERE A WALLY IN AN IMAGE?

To study the optimization and generalization properties of CNNs, we build two datasets: one derived from the MNIST (Lecun et al., 1998) dataset and another one produced by cropping large resolution images from the CAMELYON dataset (Ehteshami Bejnordi et al., 2017). Each dataset allows to evaluate the behaviour of a CNN-based binary classifier when altering different data-related factors

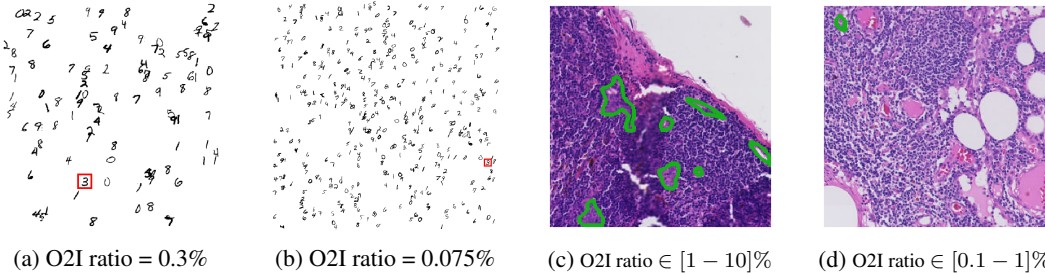

(a) O2I ratio = 0.3%   (b) O2I ratio = 0.075%   (c) O2I ratio $\in [1-10]\%$   (d) O2I ratio $\in [0.1-1]\%$

Figure 2: Example images from our nMNIST (a, b) and nCAMELYON (c, d) datasets with different O2I ratios. The object of interest is marked by red (nMNIST) or green outlines (nCAMELYON).

of variation such as dataset size, object size, image resolution and class balance. In this subsection, we describe the data generation process.

**Digits: needle MNIST (nMNIST).** Inspired by the cluttered MNIST dataset (Ba et al., 2015), we introduce a scaled up, large resolution cluttered MNIST dataset, suitable for binary image classification. In this dataset, images are obtained by randomly placing a varying number of MNIST digits on a large resolution image canvas. We keep the original $28 \times 28$ pixels digit resolution and control the O2I ratio by increasing the resolution of the canvas [1]. As result, we obtain the following O2I ratios $\{19.1, 4.8, 1.2, 0.3,$ and $0.075\}\%$ that correspond to the following canvas resolutions $64 \times 64, 128 \times 128, 256 \times 256, 512 \times 512,$ and $1024 \times 1024$ pixels, respectively. As object of interest, we select digit 3. All positive images contain exactly one instance of the digit 3 randomly placed within the image canvas, while negative instances do not contain any instance. We also include distractors (clutter digits): any MNIST digit image sampled with replacement from a set of labels $\{0, 1, 2, 4, 5, 6, 7, 8, 9\}$. We maintain approximately constant clutter density over different O2I ratios. Thus, the following O2I ratios $\{19.1, 4.8, 1.2, 0.3,$ and $0.075\}\%$ correspond to $2, 5, 25, 100,$ and $400$ clutter objects, respectively. For each value of O2I ratio, we obtain $11276, 1972, 4040$ of training, validation and test images[2]. Fig. 2 depicts example images for different O2I ratios. We refer the interested reader to the supplementary material for details on image generation process as well as additional dataset visualizations.

**Histopathology: needle CAMELYON (nCAMELYON).** The CAMELYON (Ehteshami Bejnordi et al., 2017) dataset contains gigapixel hystopathology images with pixel-level lesion annotations from 5 different acquisition sites. We use the pixel-wise annotations to extract crops with controlled O2I ratios. Namely, we generate datasets for O2I ratios in the range of $(100-50)\%, (50-10)\%, (10-1)\%,$ and $(1-0.1)\%$, and we crop different image resolutions with the size of $128 \times 128, 256 \times 256,$ and $512 \times 512$ pixels. This results in training sets of about $20-235$ unique lesions per dataset configuration (see supplementary for a detailed list of dataset sizes). More precisely, positive examples are created by taking 50 random crops from every contiguous lesion annotation and rejecting the crop if the O2I ratio does not fall within the desired range. Negative images are taken by randomly cropping healthy images and filtering image crops that mostly contain background. We ensure the class balance by sampling an equal amount of positive and negative crops. Once the crops were extracted, no pixel-wise information is used during training. Figure 2 shows examples of extracted images used in the nCAMELYON dataset experiments. We refer to the supplementary for more detail about the data extraction process, the resulting dataset sizes and more visualizations.

## 2.2 MODELS

Our classification pipelines follow BagNets (Brendel & Bethge, 2019) backbone, which allows us to explicitly control for the network receptive field size. Figure 3 shows a schematic of our approach. As can be seen, the pipelines are built of three components: (1) topological embedding extractor in

---

[1]Alternatively, we could fix canvas image resolution and downscale MNIST digits; however, downscaling might reduce the object quality.

[2]We obtain those numbers by using the original MNIST data, we use every digit 3 only once to generate positive images and we balance the dataset with negative images. See supplementary material for class imbalanced data scenarios.

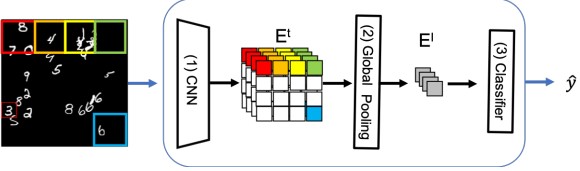

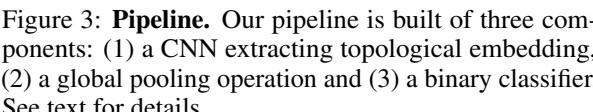

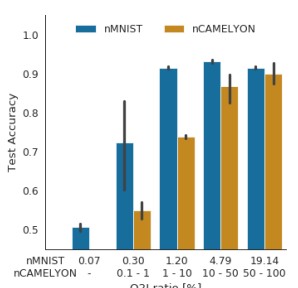

Figure 3: **Pipeline.** Our pipeline is built of three components: (1) a CNN extracting topological embedding, (2) a global pooling operation and (3) a binary classifier. See text for details.

Figure 4: **Image-level annotations**. Test set accuracy vs. O2I ratio for best models, see text for details.

which we can control for embedding receptive field, (2) global pooling operation that converts the topological embedding into a global embedding, and (3) a binary classifier that receives the global embedding and outputs binary classification probabilities. By varying the embedding extractor and the pooling operation, we test a set of 48 different architectures.

**Topological embedding extractor.** The extractor takes as input an image $\mathbf{I}$ of size $[w_{img} \times h_{img} \times c_{img}]$ and outputs a topological embedding $\mathbf{E}^t$ of shape $[w_{enc} \times h_{enc} \times c_{enc}]$, where $w_.$, $h_.$, and $c_.$ represent width, height and number of channels. Due to the relatively large image sizes, we train the pipeline with small batch sizes and, thus, we replace BagNet-used BatchNorm operation (Ioffe & Szegedy, 2015) with Instance Normalization (Ulyanov et al., 2016). In our experiments, we test 12 different extractor architectures obtained by adapting embedding extractor receptive field and capacity. For model details, please refer to section B.2 in the supplementary material.

**Global pooling operation.** Global pooling operation takes as an input topological embedding $\mathbf{E}^t$ of shape $[w_{enc} \times h_{enc} \times c_{enc}]$ and outputs global image embedding $\mathbf{E}^I$ of shape $[1 \times 1 \times c_{enc}]$. In the paper, we experiment with four different pooling operations, namely: max, logsumexp, average, and soft attention. In our experiments, we follow the soft attention formulation of (Ilse et al., 2018). The details about global pooling operations can be found in the supplementary material.

## 3 EXPERIMENTAL RESULTS

In this section, we experimentally test how CNNs' optimization and generalization scale with *low* and *very low* O2I ratios. First, we provide details about our experimental setup and then we design experiments to provide empirical evidence to the following questions: (1) **Image-level annotations**: Is it possible to train classification systems that generalize well in low and very low O2I scenarios? (2) **O2I limit vs. dataset size**: Is there an O2I ratio limit below which the CNNs will experience generalization difficulties? Does this O2I limit depend on the dataset size? (3) **O2I limit vs. model capacity**: Do higher capacity models generalize better? (4) **Inductive bias - receptive field**: Is adjusting receptive field size to match (or exceed) the expected object size beneficial? (5) **Global pooling operations**: Does the choice of global pooling operation affect model generalization? Finally, we inquire about the **optimization** ease of the models trained on data with very low O2I ratios.

In all our experiments, we used RMSProp (Tieleman & Hinton, 2012) with a learning rate of $\eta = 5 \cdot 10^{-5}$ and decayed the learning rate multiplying it by 0.1 at 80, 120 and 160 epochs [3]. All models were trained with cross entropy loss for a maximum of 200 epochs. We used an effective batch size of 32. If the batch did not fit into memory we used smaller batches with gradient accumulation. To ensure robustness of our conclusions, we run every experiment with six different random seeds and report the mean and standard deviation. Throughout the training we monitored validation accuracy, and reported test set results for the model that achieved best validation set performance.

---

[3]Before committing to a single optimization scheme, we evaluated a variety of optimizers (Adam, RMSprop and SGD with momentum), learning rates ($\eta \in \{1, 2, 3, 5, 7, 10\} \cdot 10^{-5}$), and 3 learning rate schedules.

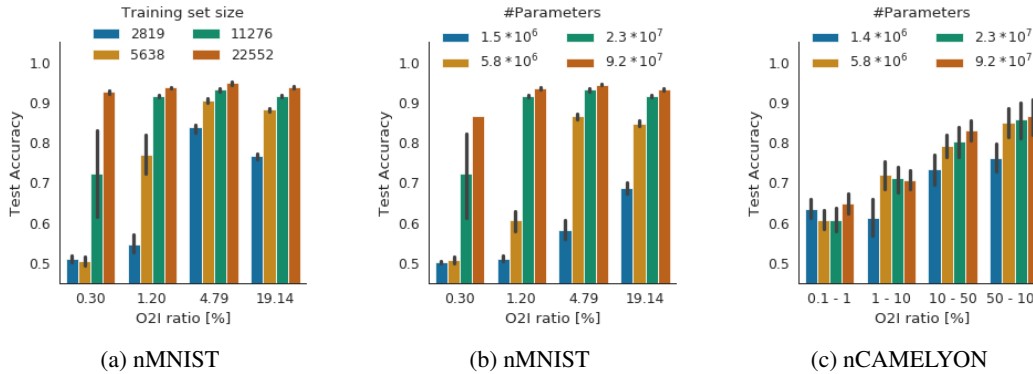

(a) nMNIST          (b) nMNIST          (c) nCAMELYON

Figure 5: **Testing the O2I limit.** Subfigure (a) depicts the test set performance as a function of training dataset size for the nMNIST dataset, while subfigures (b) and (c) show the test set performance as a function of model capacity for the nMNIST dataset and the nCAMELYON dataset, respectively.

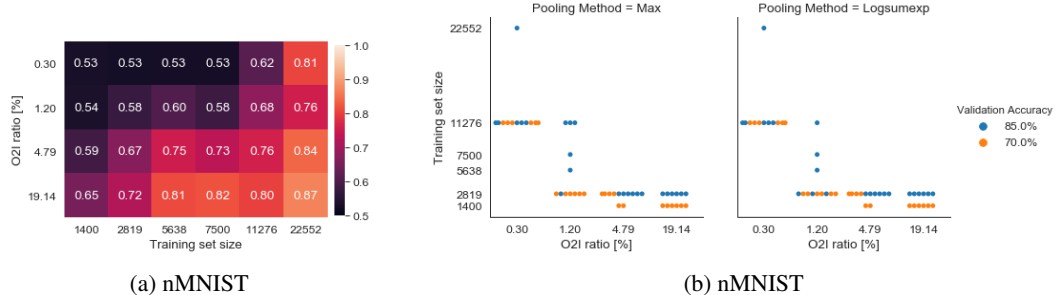

(a) nMNIST                              (b) nMNIST

Figure 6: **Testing the O2I limit.** (a) mean validation set accuracy heatmap for max pooling operation, and (b) minimum required training set size to achieve the noted validation accuracy. We test training set sizes $\in \{1400, 2819, 5638, 7500, 11276, 22552\}$ and report the minimum amount of training examples that achieve a specific validation performance pooling over different network capacities.

### 3.1 RESULTS

In this subsection, we present and discuss the main results of our analysis. Unless stated otherwise, the capacity of the ResNet-50 network is about $2.3 \cdot 10^7$ parameters. Additional results and analysis are presented in the supplementary material.

**Image-level annotations:** For this experiment, we vary the O2I ratio on nMNIST and nCAMELYON to test its influence on the generalization of the network. Figure 4 depicts the results for the best configuration according to the validation performance: we use max-pooling and receptive field sizes of $33 \times 33$ and $9 \times 9$ pixels for the nMNIST and nCAMELYON datasets, respectively. For the nMNIST dataset, the plot represents the mean over 6 random seeds together with the standard deviation; while for the nCAMELYON dataset we report an average over both the 6 seeds and the crop sizes. We find that the tested CNNs achieve reasonable test set accuracies for the O2I ratios larger than $0.3\%$ for the nMNIST datset and the O2I ratios above $1\%$ for the histopathology dataset. For both datasets, smaller O2I ratios lead to poor or even random test set accuracies.

**O2I limit vs. dataset size:** We test the influence of the training set size on model generalization for the nMNIST data, to understand the CNNs' generalization problems for very small O2I ratios. We tested six different dataset sizes $(1400, 2819, 5638, 7500, 11276, 22552)$ [4]. Fig. 5a depicts the results for max-pooling and a receptive field of $33 \times 33$ pixels. We observe that larger datasets yield better generalization and this increment is more pronounced for small O2I ratios. For further insights, we plot a heatmap representing the mean validation set results [5] for all considered 02Is and training set sizes (Fig. 6a) as well as the minimum number of training examples to achieve a validation accuracy

---

[4] We allow to reuse each digit 3 for larger training sets and select a subset for smaller training sets.

[5] More precisely, we plot the mean of all pipeline configurations that surpassed $70\%$ training accuracy.

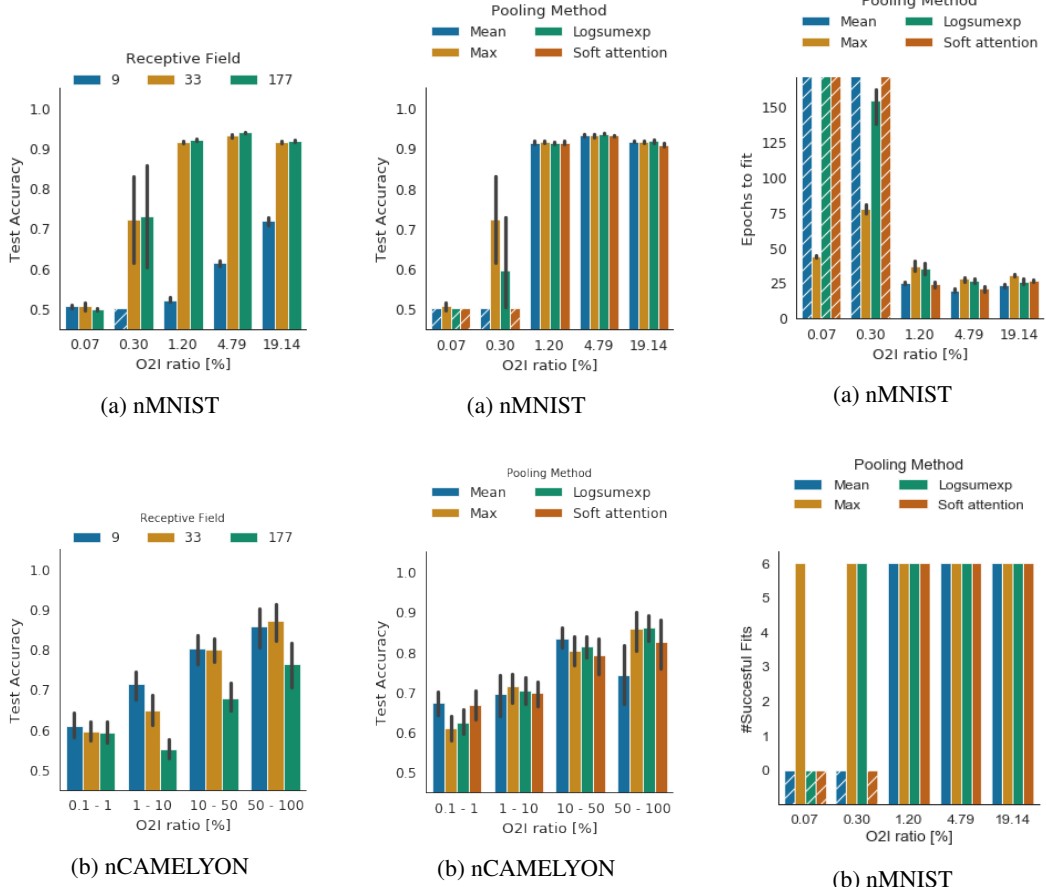

(a) nMNIST  (a) nMNIST  (a) nMNIST

(b) nCAMELYON  (b) nCAMELYON  (b) nMNIST

Figure 7: **Inductive bias:** for (a) the nMNIST dataset and (b) the nCAMELYON dataset. We report only runs that fit the training data. Otherwise we report random accuracy and depict it with a texture on the bars.

Figure 8: **Global pooling operations:** for (a) the nMNIST dataset and (b) the nCAME-LYON datset. We report only runs that fit the training data. Otherwise we report random accuracy and depict it with a texture on the bars.

Figure 9: **nMNIST optimization:** (a) number of training epochs needed to fit the 11k training data and (b) the number of successful runs. The textured bars indicate that the model did not fit the training data for all random seeds.

of 70% and 85% (Fig. 6b). We observe that in order to achieve good classification generalization the required training set size rapidly increases with the decrease of the O2I ratio.

**O2I limit vs. capacity:** In this experiment, we train networks with different capacities — by uniformly scaling the initial number of filters in convolutional kernels by $\left[\frac{1}{4}, \frac{1}{2}, 1, \text{ and } 2\right]$[6]. We show the CNNs test set performances as a function of the O2I ratio and the network capacity in Figures 5b and 5c for the nMNIST (with 11k training points) and nCAMELYON data, respectively. On nMNIST, we observe a clear trend, where the model test set performance increases with capacity and this boost is larger for smaller O2Is. We hypothesize, that this generalization improvement is due to the model ability to learn-to-ignore the input data noise; with smaller O2I there is more noise to ignore and, thus, higher network capacity is required to solve the task. However, for the nCAMELYON dataset, this trend is not so pronounced and we attribute this to the limited dataset size (more precisely to the small number of unique lesions). These results suggest that collecting a very large histopathology dataset might enable training of CNN models using *only* image level annotations.

**Inductive bias - receptive field:** We report the test accuracy as a function of the O2I ratio and the receptive field size for nMINIST in Figure 7a and for nCAMELYON in Figure 7b. Both plots depict

---

[6]We chose the maximum scaling factor so that the largest resolution images still fit in available GPU memory. For images with O2I ratio of 0.07, the available GPU memory prevents testing networks with higher capacity.

results for the global max pooling operation. For nMNIST, we observe that a receptive field that is bigger than the area occupied by one single digit leads to best performances; for example, receptive fields of $33 \times 33$ and $177 \times 177$ pixels clearly outperform the smallest tested receptive field of $9 \times 9$ pixels. However, for the nCAMELYON dataset we observe that the smallest receptive field actually performs best. This suggests that most of the class-relevant information is contained in the texture and that higher receptive fields pick up more spurious correlations, because the capacity of the networks is constant.

**Global pooling operations:** In this experiment, we compare the performance of four different pooling approaches. We present the relation between test accuracy and pooling function for different O2I ratios with a receptive field of $33 \times 33$ pixels for nMNIST in Figure 8a and $9 \times 9$ pixels for nCAMELYON in Figure 8b. On the one hand, for the nMNIST dataset, we observe that for the relatively large O2I ratios, all pooling operations reach similar performance; however, for smaller O2Is we see that max-pooling is the best choice. We hypothesize that the global max pooling operation is best suited to remove nMNIST-type of structured input noise. On the other hand, when using the histopathology dataset, for the smallest O2I mean and soft attention poolings reach best performances; however, these outcomes might be affected by the relatively small nCAMELYON dataset used for training.

**Optimization:** In our large scale nMNIST experiments (when using $\approx$ 11k datapoints), we observed that some configurations have problems fitting the training data [7]. In some runs, after significant efforts put into CNNs hyperparamenter selection, the training accuracy was close to random. To investigate this issue further, we followed the setup of randomized experiments from (Zhang et al., 2017; Arpit et al., 2017) and we substituted the nMNIST datapoints with samples from an isotropic Gaussian distribution. On the one hand, we observed that all the tested setups of our pipeline were able to memorize the Gaussian samples, while, on the other hand, most setups were failing to memorize the same-size, nMNIST datataset for small and very small O2I ratios. We argue that the nMNIST structured noise and its compositionality may be a "harder" type of noise for the CNNs than Gaussian isotropic noise. To provide further experimental evidence, we depict average time-to-fit the training data (in epochs) in Fig. 9a as well as number of successful optimizations in Fig. 9b for different O2I ratios and pooling methods[8]. We observe that the optimization gets progressively harder with decreasing O2I ratio (with max pooling being the most robust). Moreover, we note that the results are consistent across different random seeds, where all runs either succeed or fail to converge.

# 4 RELATED WORK

## 4.1 TINY OBJECT CLASSIFICATION

Reasoning about tiny objects is of high interest in many computer vision areas, such as medical imaging (Ehteshami Bejnordi et al., 2017; Aresta et al., 2018; Setio et al., 2017; Suckling, 1994; Sudre et al., 2018) and remote sensing (Xia et al., 2018; Pang et al., 2019). To overcome the low signal-to-noise ratio, most approaches rely on manual dataset "curation" and collect additional *pixel-level annotations* such as landmark positions (Borovec et al., 2018), bounding boxes (Wei et al., 2019; Resta et al., 2011) or segmentation maps (Ehteshami Bejnordi et al., 2017). This additional annotation allows to transform the original needle-in-a-haystack problem into a less noisy but imbalanced classification problem (Wei et al., 2019; Lee & Paeng, 2018; Bándi et al., 2019). However, collecting pixel level annotations has a significant cost and might require expert knowledge, and as such, is a bottleneck in the data collection process.

Other approaches leverage the fact that task-relevant information is often not uniformly distributed across input data, e.g. by using attention mechanisms to process very high-dimensional inputs (Mnih et al., 2014; Ba et al., 2015; Almahairi et al., 2016; Katharopoulos & Fleuret, 2019). However, those approaches are mainly motivated from a computational perspective trying to reduce the computational footprint at inference time.

---

[7]We did not observe optimization problems for small dataset sizes of the nMNIST nor for nCAMELYON.

[8]We define an optimization to be successful if it the training set accuracy surpassed 99%.

Some recent research has also studied attention based approaches both in the context of multi-instance learning (Ilse et al., 2018) and histopathology image classification (Tomita et al., 2018). However, neither of the works report the exact O2I ratio used in the experiments.

## 4.2 GENERALIZATION OF CNNS

In this subsection, we briefly highlight the dimensions of optimization and generalization of CNN that are handy in low O2I classification scenarios.

**Model capacity.** For fixed training accuracy, over-parametrized CNNs tend to generalize better (Novak et al., 2018). In addition, when properly regularized and given a fixed size dataset, higher capacity models tend to provide better performance (He et al., 2016; Huang et al., 2017). However, finding proper regularization is not trivial (Goodfellow et al., 2016).

**Dataset size.** CNN performance improves logarithmically with dataset size (Sun et al., 2017). Moreover, in order to fully exploit the data benefit, the model capacity should scale jointly with the dataset size (Mahajan et al., 2018; Sun et al., 2017).

**Model inductive biases.** Inductive biases limit the space of possible solutions that a neural network can learn (Goodfellow et al., 2016). Incorporating these biases is an effective way to include data (or domain) specific knowledge in the model. Perhaps the most successful inductive bias is the use of convolutions in CNNs (LeCun & Bengio, 1998). Different CNN architectures (e. g. altering network connectivity) also lead to improved model performance (He et al., 2016; Huang et al., 2017). Additionally, it has been shown on the ImageNet dataset that CNN accuracy scales logarithmically with the size of the receptive field (Brendel & Bethge, 2019).

## 5 DISCUSSION AND CONCLUSIONS

Although low input image signal-to-noise scenarios have been extensively studied in signal processing field (e.g. in tasks such as image reconstruction), less attention has been devoted to low signal-to-noise classification scenarios. Thus, in this paper we identified an *unexplored* machine learning problem, namely image classification in *low* and *very low* signal-to-noise ratios. In order to study such scenarios, we built two datasets that allowed us to perform controlled experiments by manipulating the input image signal-to-noise ratio and highlighted that CNNs struggle to show good generalization for *low* and *very low* signal-to-noise ratios even for a relatively elementary MNIST-based dataset. Finally, we ran a *series of controlled experiments*[9] that explore both a variety of CNNs' architectural choices and the importance of training data scale for the *low* and *very low* signal-to-noise classification. One of our main observation was that properly designed CNNs can be *trained in low O2I regime without using any pixel-level annotations and generalize* if we leverage enough training data; however, *the amount of training data required for the model to generalize scales rapidly with the inverse of the O2I ratio*. Thus, with our paper (and the code release) we invite the community to work on data-efficient solutions to *low* and *very low* signal-to-noise classification.

Our experimental study exhibits limitations: First, due to the lack of large scale datasets that allow for explicit control of the input signal-to-noise ratios, we were forced to use the synthetically built nMNIST dataset for most of our analysis. As a real life dataset, we used crops from the histopathology CAMELYON dataset; however, due to relatively a small number of unique lesions we were unable to scale the histopathology experiments to the extent as the nMNIST experiments, and, as result, some conclusions might be affected by the limited dataset size. Other large scale computer vision datasets like MS COCO (Lin et al., 2014) exhibit correlations of the object of interest with the image background. For MS COCO, the smallest O2I ratios are for the object category "sports ball" which on average occupies between $0.3\%$ and $0.4\%$ of an image and its presence tends to be correlated with the image background (e. g. presence of sports fields and players). However, future research could examine a setup in which negative images contain objects of the categories "person" and "baseball bat" and positive images also contain "sports ball". Second, all the tested models improve the generalization with larger dataset sizes; however, scaling datasets such as CAMELYON to tens of thousands of samples might be prohibitively expensive. Instead, further research should be devoted to developing computationally-scalable, data-efficient inductive biases that can handle very low

---

[9]We ran more than 750 experiments each with 6 different seeds.

signal-to-noise ratios with limited dataset sizes. Future work, could explore the knowledge of the low O2I ratio and therefore sparse signal as an inductive bias. Finally, we studied low signal-to-noise scenarios only for binary classification scenarios [10]; further investigation should be devoted to multi-class problems. We hope that this study will stimulate the research in image classification for low signal-to-noise input scenarios.

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

# Supplementary Material for
# Needles in Haystacks: On Classifying Tiny Objects in Large Images

## A DATASETS

In this section, we provide additional details about the datasets used in our experiments.

### A.1 NEEDLE MNIST

The needle MNIST (nMNIST) dataset is designed as a binary classification problem: *Is there a 3 in this image?'*. To generate nMINST, we use the original training, validation and testing splits of the MNIST dataset and generate different nMINST subsets by varying the object-to-image (O2I) ratio, resulting in O2I ratios of $19.1\%, 4.8\%, 1.2\%, 0.3\%,$ and $0.075\%$. We define *positive* images as the ones containing exactly one digit 3 and *negative* images as images without any instance of it. We keep the original MNIST digit size and place digits randomly onto a clear canvas to generate a sample of the nMNIST dataset. More precisely, we adapt the O2I ratio by changing the the canvas size, resulting in nMNIST image resolution being in $64 \times 64, 128 \times 128, 256 \times 256, 512 \times 512,$ and $1024 \times 1024$ pixels. To assign MNIST digits to canvas, we split the MNIST digits into two subsets: *digit-3* versus *clutter* (any digit from a set of $\{0, 1, 2, 4, 5, 6, 7, 8, 9\}$). For the positive nMNIST images, we sample one digit 3 (without replacement) and $n$ digits (with replacement) from the digit-3 and clutter subsets, respectively. For the negative nMNIST images, we sample $n + 1$ instances from the clutter subset. We adapt $n$ to keep approximately constant object density for all canvas and choose $n$ to be $2, 5, 25, 100,$ and $400$ for canvas resolutions $64 \times 64, 128 \times 128, 256 \times 256, 512 \times 512,$ and $1024 \times 1024$, respectively. As result, for each value of O2I ratio, we obtain $11276, 1972, 4040$ of training, validation and testing images, out of which $50\%$ are negative and $50\%$ are positive images. We present both positive and negative samples for different O2I ratios in Figure 10.

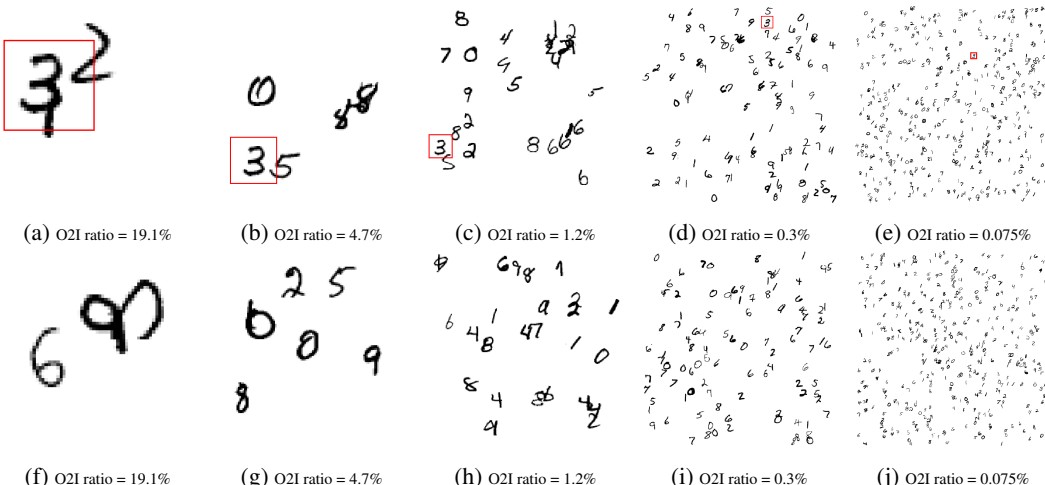

| (a) O2I ratio = 19.1% | (b) O2I ratio = 4.7% | (c) O2I ratio = 1.2% | (d) O2I ratio = 0.3% | (e) O2I ratio = 0.075% |

| (f) O2I ratio = 19.1% | (g) O2I ratio = 4.7% | (h) O2I ratio = 1.2% | (i) O2I ratio = 0.3% | (j) O2I ratio = 0.075% |

Figure 10: Example images from our MNIST dataset with different O2I ratios. Top row images represent positive examples — digit 3 is present (marked with red rectangle), while bottom row depicts negative images. Note that for visualization purposes all images have been rescaled to the same resolution.

### A.2 NEEDLE CAMELYON

The needle CAMELYON (nCAMELYON) is designed as a binary classification task: *Are there breast cancer metastases in the image or not?*. We rely on the pixel-level annotations within CAMELYON to extract samples for nCAMELYON. We use downsampling level 3 from the original whole slide image using the MultiResolution Image interface released with the original CAMELYON dataset. For positive examples, we identify contiguous regions within the annotations, and take 50 random crops around each contiguous region ensuring that the full contiguous region is inside the crop, and

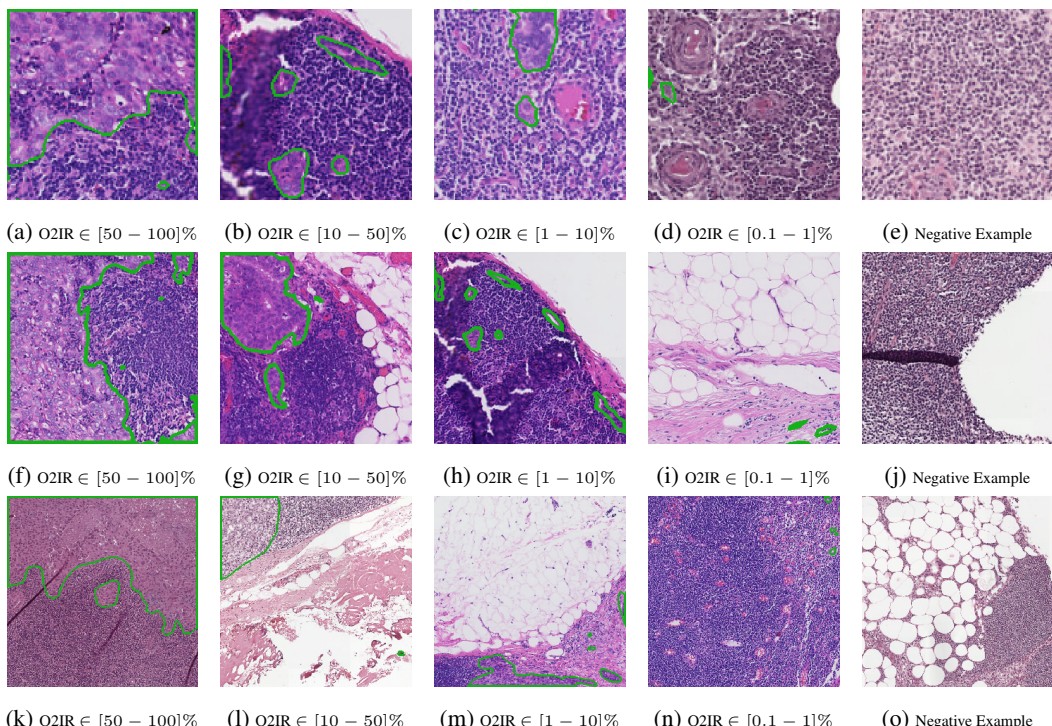

(a) O2IR ∈ [50 − 100]%  (b) O2IR ∈ [10 − 50]%  (c) O2IR ∈ [1 − 10]%  (d) O2IR ∈ [0.1 − 1]%  (e) Negative Example

(f) O2IR ∈ [50 − 100]%  (g) O2IR ∈ [10 − 50]%  (h) O2IR ∈ [1 − 10]%  (i) O2IR ∈ [0.1 − 1]%  (j) Negative Example

(k) O2IR ∈ [50 − 100]%  (l) O2IR ∈ [10 − 50]%  (m) O2IR ∈ [1 − 10]%  (n) O2IR ∈ [0.1 − 1]%  (o) Negative Example

Figure 11: Example images from our CAMELYON dataset for different crop sizes and O2I ratios. We show crops with size $128 \times 128$, $256 \times 256$, and $512 \times 512$ in the top, middle, and bottom row, respectively. The green outlines show the cancerous regions. Note that for visualization purposes all images have been rescaled to same resolution.

total number of lesion pixels inside the crop are in the desired O2I ratio. The negative crops are taken from healthy images randomly filtering for images that are mostly background using a heuristic that the average green pixel value in the crop is below 200. Since the CAMELYON dataset contains images acquired by 5 different centers, we split training, validation and test sets center-wise to avoid any contamination of data across the three sets. All crops coming from center 3 are part of the validation set, and all crops coming from center 4 are part of the test set. All images are generated for resolutions $128 \times 128$, $256 \times 256$, $512 \times 512$, and $1024 \times 1024$ and are split into 4 different O2I ratios: $(100 − 50)\%$, $(50 − 10)\%$, $(10 − 1)\%$, and $(1 − 0.1)\%$. Figure 11 shows examples of images from nCAMELYON dataset, Table 1 presents number of unique lesions in each dataset, and Table 2 depicts number of dataset images stratified for image resolution and O2I ratios. Because center 3 does not contain lesions of suitable size for crops of with resolution $128 \times 128$ and O2I ratio $(50 − 100)\%$, we do not include those training runs in our analysis.

Table 1: Number of unique lesions extracted for each set of the nCAMELYON data for differen O2I ratios and crop sizes.

| O2I ratio | Crop Size | 128 | | | 256 | | | 512 | | |
|---|---|---|---|---|---|---|---|---|---|---|
| | | Train | Val | Test | Train | Val | Test | Train | Val | Test |
| (50 - 100)% | | 20 | 0 | 8 | 27 | 2 | 13 | 23 | 5 | 13 |
| (10 - 50)% | | 84 | 12 | 16 | 101 | 16 | 15 | 68 | 15 | 17 |
| (1 - 10)% | | 176 | 17 | 18 | 227 | 17 | 18 | 235 | 21 | 15 |
| (0.1 - 1)% | | 33 | 5 | 5 | 93 | 16 | 9 | 173 | 20 | 11 |

Table 2: Number of crops extracted for each set of the nCAMELYON data for differen O2I ratios and crop sizes. Note that the dataset is balanced (e. g. $50\%$ are positive images and $50\%$ are negative). Moreover, for positive images we have relatively small number of unique cancer regions as noted in Table 1.

| Crop Size | 128 | | | 256 | | | 512 | | |
|---|---|---|---|---|---|---|---|---|---|
| O2I ratio | Train | Val | Test | Train | Val | Test | Train | Val | Test |
| (50 - 100)% | 1000 | 0 | 400 | 1350 | 100 | 650 | 1150 | 250 | 650 |
| (10 - 50)% | 4200 | 600 | 800 | 5050 | 800 | 750 | 3400 | 750 | 850 |
| (1 - 10)% | 8686 | 850 | 900 | 11270 | 850 | 900 | 11750 | 1050 | 750 |
| (0.1 - 1)% | 1488 | 247 | 207 | 4255 | 800 | 450 | 8312 | 965 | 550 |
| negative | 19608 | 6000 | 6100 | 19595 | 6000 | 6100 | 19574 | 6000 | 6100 |

## B    EXPERIMENTAL SETUP

In this section, we provide additional details about the pipeline used in the experiments. More precisely, we formally define global pooling operations and provide detailed description of the different architectures.

### B.1    GLOBAL POOLING OPERATIONS

In our experiments, we are testing four different global pooling functions: max-pooling, mean-pooling, logsumexp and soft attention. The max pooling operation simply returns the maximum value per each channel in the topological embedding. This operation can be formally defined as: $\mathbf{E}^I = \max_w \max_h \mathbf{E}^t_{[w,h]}$. Note, that we use subscript notation to denote dimensions of the embedding. The max pooling operation has a spacing effect on gradient backpropagation, during the backward pass through the model all information will be propagated through the embedding position that corresponds to the maximal value. In order to improve gradient backpropagation, one could apply logsumexp pooling, a soft approximation to max pooling. This pooling operation is defined as:

$$\mathbf{E}^I = \log \sum_{w=1}^{w_{enc}} \sum_{h=1}^{h_{enc}} \exp \mathbf{E}^t_{[w,h]}. \tag{1}$$

Alternatively, one could use an average pooling operation that computes mean value for each channel in the topological embedding. This pooling operation can be formally defined as follows:

$$\mathbf{E}^I = \frac{1}{w_{enc}} \frac{1}{h_{enc}} \sum_{w=1}^{w_{enc}} \sum_{h=1}^{h_{enc}} \mathbf{E}^t_{[w,h]}. \tag{2}$$

Finally, attention based pooling include additional weighting tensor $\mathbf{a}$ of dimension $(w_{enc} \times h_{enc} \times c_{enc})$ that rescales each topological embedding before averaging them. This operation can be formally defined as:

$$\mathbf{E}^I = \sum_{w=1}^{w_{enc}} \sum_{h=1}^{h_{enc}} a_{[w,h]} \cdot \mathbf{E}^t_{[w,h]} \tag{3}$$

$$s.t. \sum_{w=1}^{w_{enc}} \sum_{h=1}^{h_{enc}} \mathbf{a}_{[w,h]} = 1 \tag{4}$$

In our experiments, following Ilse et al. (2018), we parametrize the soft-attention mechanisms as $\mathbf{a}_{[w,h]} = softmax(f(\mathbf{E}_{spat}))_{[w,h]}$, where $f(\cdot)$ is modelled by two fully connected layers with tanh-activation and 128 hidden units.

### B.2    MODEL ARCHITECTURE DETAILS

We adapt the BagNet architecture proposed in (Brendel & Bethge, 2019). An overview of the architectures for the tested three receptive field sizes is shown in Table 3. We depict the layers of

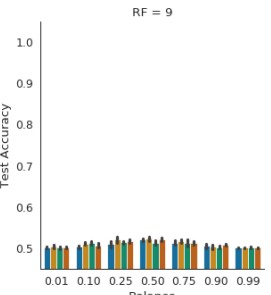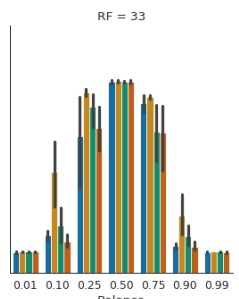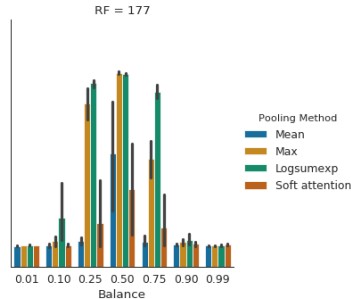

Figure 12: Impact of the training set balance on model accuracy for different pooling operations and receptive field sizes.

residual blocks in brackets and perform downsampling using convolutions with stride 2 within the first residual block. Note that the architectures for different receptive fields differ in the number of $3 \times 3$ convolutions. The rightmost column shows a regular ResNet-50 model. The receptive field is decreased by replacing $3 \times 3$ convolutions with $1 \times 1$ convolutions. We increase the number of convolution filters by a factor of $2.5$ if the receptive field is reduced to account for the loss of the trainable parameters. Moreover, when testing different network capacities we evenly scale the number of convolutional filters by multiplying with a constant factor of $s \in \{1/4, 1/2, 1, 2\}$.

## C  ADDITIONAL RESULTS

In this section, we provide additional experimental results as well as additional visualizations of the experiments presented in the main body of the paper.

### C.1  CLASS-IMBALANCED CLASSIFICATION

In many medical imaging datasets, it is common to be faced with class-imbalanced datasets. Therefore, in this experiment, we use our nMNIST dataset and test CNNs generalization under moderate and severe class imbalanced scenario. We alter the training set class balance by altering the proportion of positive images in the training dataset and use the following balance values $0.01$, $0.1$, $0.25$, $0.5$, $0.75$, $0.9$ and $0.99$, where a value of $0.01$ means almost no positive examples and $0.99$ indicates very low number of negative images available at training time. Moreover, we ensure that the dataset size is constant ($\approx 11k$) and only the class-balance is modified. We run the experiments using the O2I ratio of $1.2\%$, three receptive field sizes ($9 \times 9$, $33 \times 33$ and $177 \times 177$ pixels) and four pooling operations (mean, max, logsumexp and soft attention). For each balance value, we train 6 models using 6 random seeds and we oversample the underrepresented class. The results are depicted in Figure 12. We observe that the model performance drops as the the training data becomes more unbalanced and that max pooling and logsumexp seem to be the most robust to the class imbalance.

### C.2  INCREASE OF MODEL CAPACITY FOR SMALL DATASET SIZES.

We also tested the effect of model capacity increase while having access only to a small dataset (3k class-balanced images) and contrast it with a larger dataset of $\approx 11k$ training images. We run this experiment on the nMNIST dataset using a network with $2.3 \cdot 10^7$ parameters using global max pooling operation and there different receptive field sizes: $9 \times 9$, $33 \times 33$ and $177 \times 177$ pixels. The results are depicted in Figure 13. It can be seen that the model's capacity increase does not lead to better generalization, for small size datasets of $\approx 3k$.

### C.3  O2I LIMIT VS. DATASET SIZE

In this section, we report additional results for all tested global pooling operations on O2I limit vs. dataset size. We plot a heatmaps representing the validation set results for all considered 02I and training set sizes (Figure 14) as well as the minimum number of training examples required to achieve a validation accuracy of $70\%$ and $85\%$ (Figure 15)

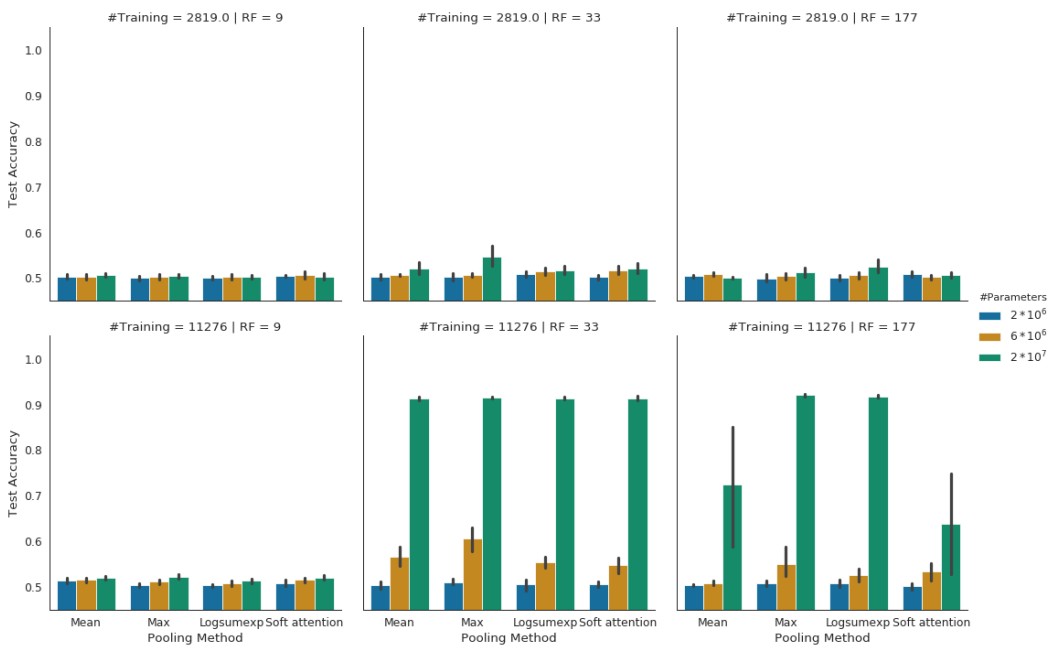

Figure 13: Impact of the network capacity on the generalization performance dependent on the training set size for nMNIST at O2I ratio = 1.2%. The improvement based on the increased network capacity shrinks with smaller training set.

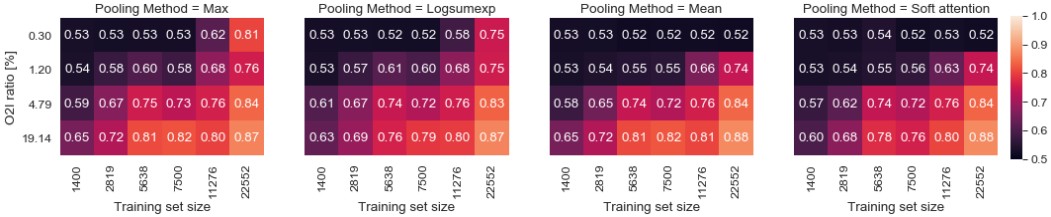

Figure 14: **Testing the O2I limit.** Validation set accuracy heatmap for max, logsumexp, mean and soft attention poolings. We test training set sizes $\in \{1400, 2819, 5638, 7500, 11276, 22552\}$ and report the average validation accuracy.

### C.4 WEAKLY SUPERVISED OBJECT DETECTION: NMNIST

We test the object localization capabilities of the trained classification models by examining their saliency maps. Figure 16 shows examples of the nMNIST dataset with the object bounding box in

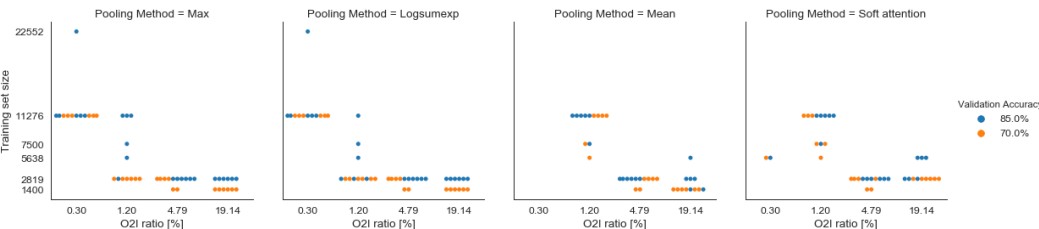

Figure 15: **Testing the O2I limit.** Minimum required training set size to achieve the noted validation accuracy. We test training set sizes $\in \{1400, 2819, 5638, 7500, 11276, 22552\}$ and report the minimum amount of training examples that achieve a specific validation performance pooling over different network capacities.

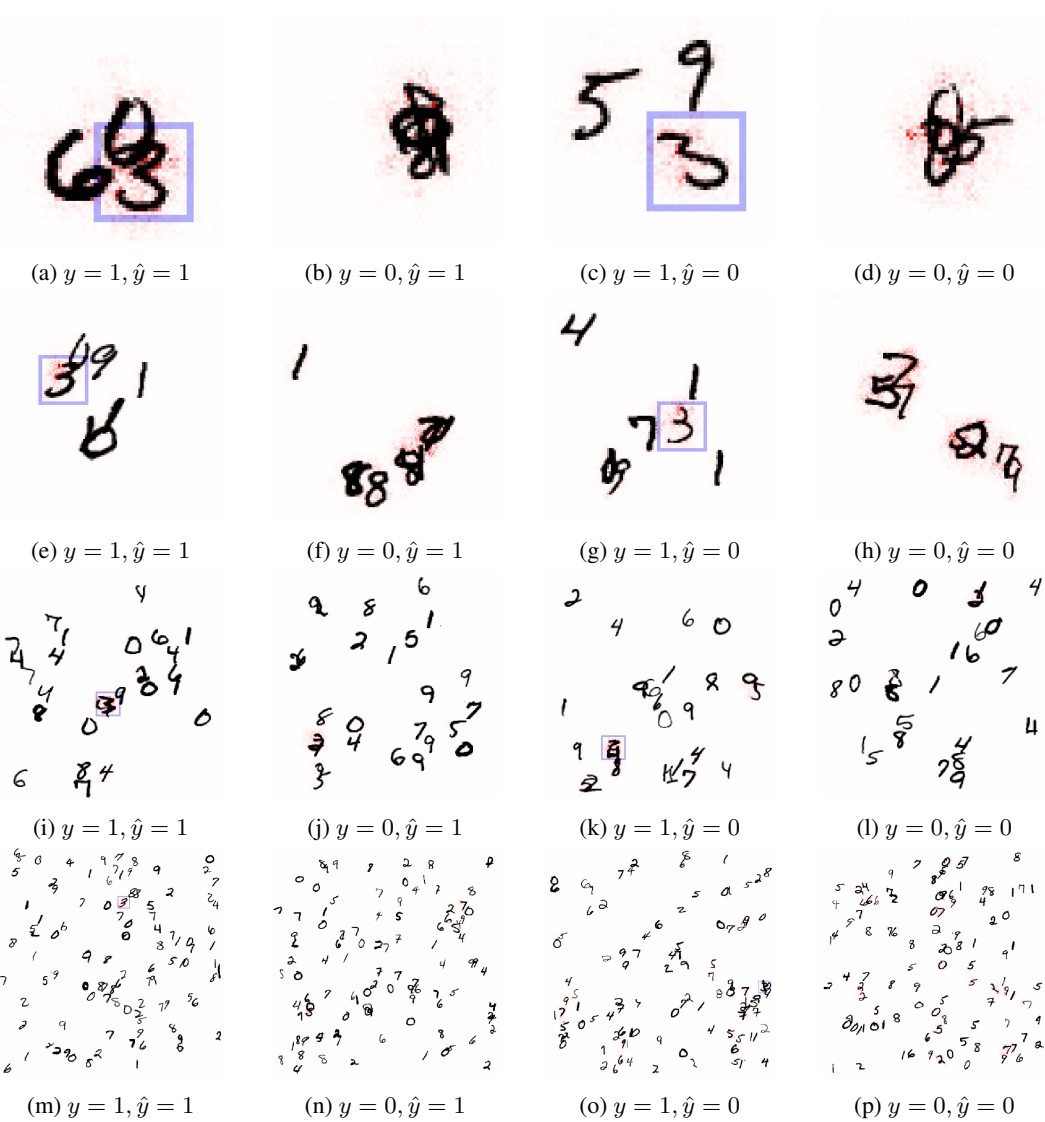

Figure 16: Example images from the nMNIST validation set and their corresponding saliency maps in red. We generate the saliency maps by calculating the absolute of the gradients with respect to the input image using max-pooling, a receptive field of 33, and ResNet-50 capacity. From top to bottom, we show random examples for O2I ratios of $\{19.14, 4.79, 1.20, 0.30\}\%$. We annotate the object of interest with a blue outline. The captions show the true label $y$ and the prediction $\hat{y}$.

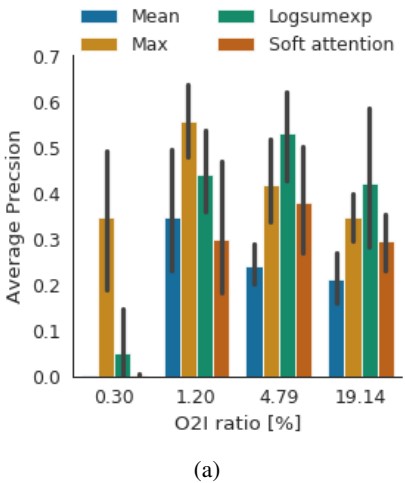 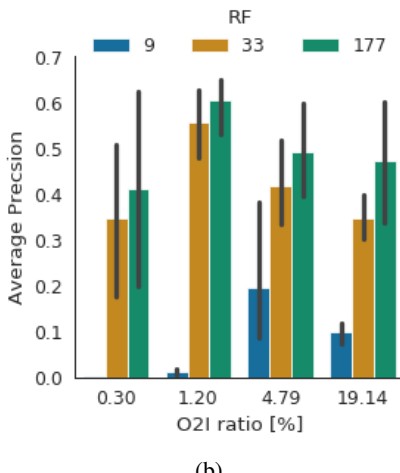

(a)                                        (b)

Figure 17: Average precision for detecting the object of interest using the saliency maps for nMNIST. We adapt (Oquab et al., 2015) and use the localize an object by the maximum magnitude of the saliency. We use the magnitude of the saliency as the confidence of the detection. We count wrongly localised objects both as false positive and false negative. For images without object of interest, the we increase the false positive count only. We plot results for max-pooling, a receptive field of 33, a training set with 11276 examples and ResNet-50 capacity. (a) shows the dependence of the AP on the pooling method using $RF = 33 \times 33$, (b) shows the dependence on the receptive field using max-pooling.

blue and the magnitude of the saliency in red. We rescale the saliency to $[0, 1]$ for better contrast. However, this prevents the comparison of absolute saliency values across different images. In samples containing an object of interest, the models correctly assign high saliency to the regions surrounding the relevant object. On negative examples, the network assigns homogenous importance to all objects.

We localise an object of interest as the location with maximum saliency. We follow (Oquab et al., 2015) to quantitatively examine the object detection performance using the saliency maps of the models. We plot the corresponding average precision in Figure 17. We find that the detection performance deteriorates for smaller O2I ratios regardless of the method. This is aligned with the classification accuracy. For small O2I ratios, max-pooling achieves the best detection scores. On larger O2I ratios, logsumexp achieves the best scores.

## C.5 WEAKLY SUPERVISED OBJECT DETECTION: NCAMELYON

We qualitatively show object detection on nCAMELYON in Figures 18 19 20 21, for True Positives, True Negatives, False Positives and False Negatives. We observe weak correlation between segmentation maps and saliency maps, signifying that the classifier was able to focus on the object of interest instead of looking at superficial signals in the data.

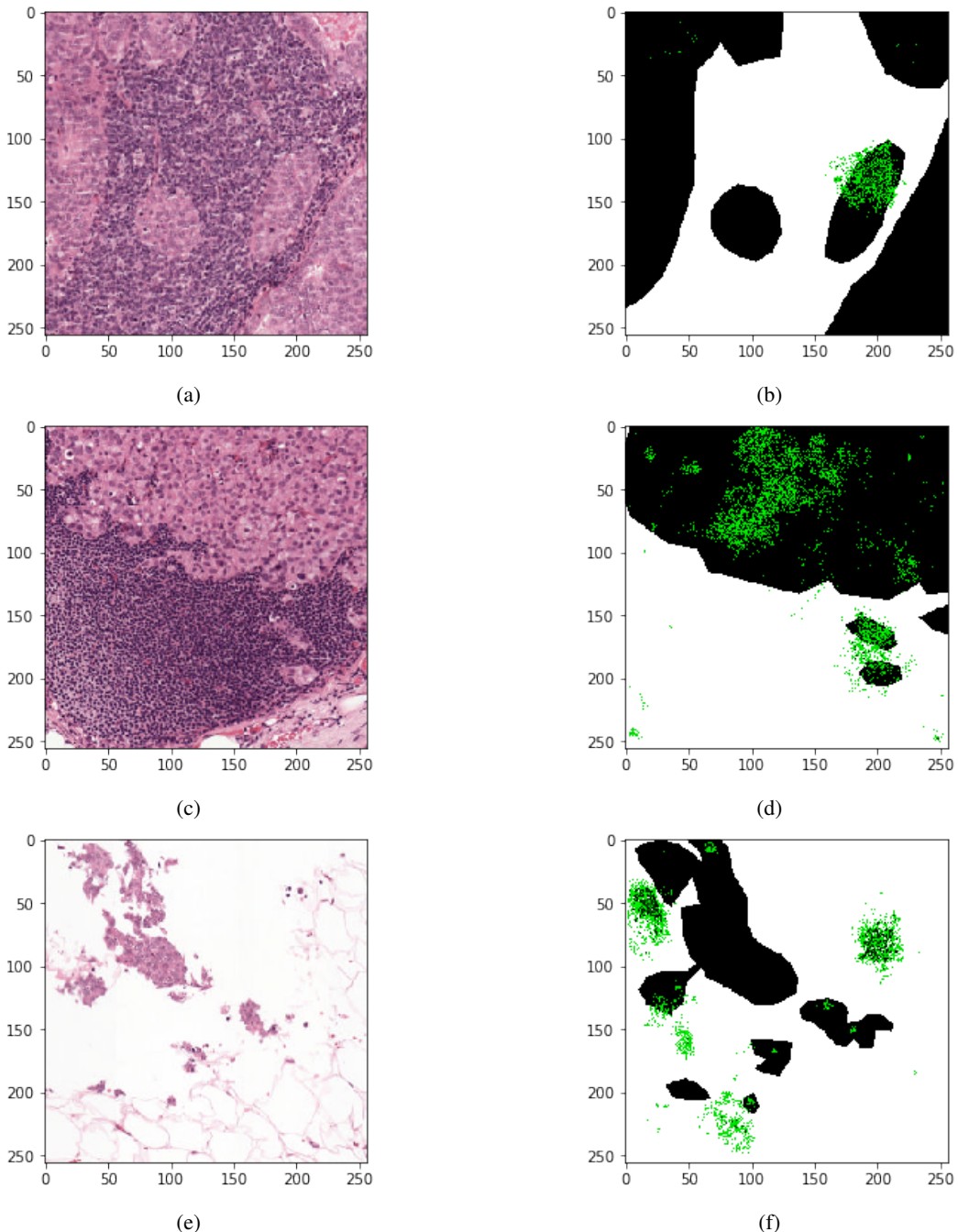

Figure 18: Example True Positive Images of nCAMELYON validation sets and their corresponding segmentation maps with saliencies overlaid.

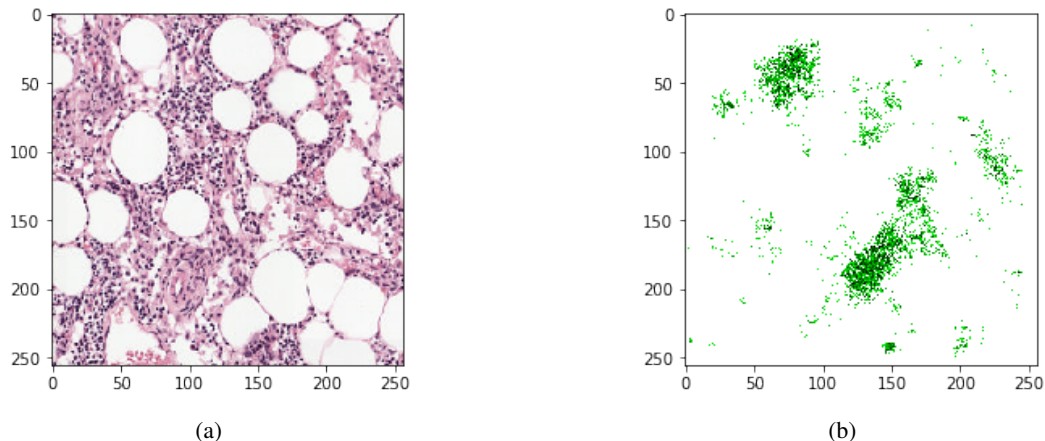

Figure 19: Example True Negative Image of nCAMELYON validation sets and corresponding saliency map.

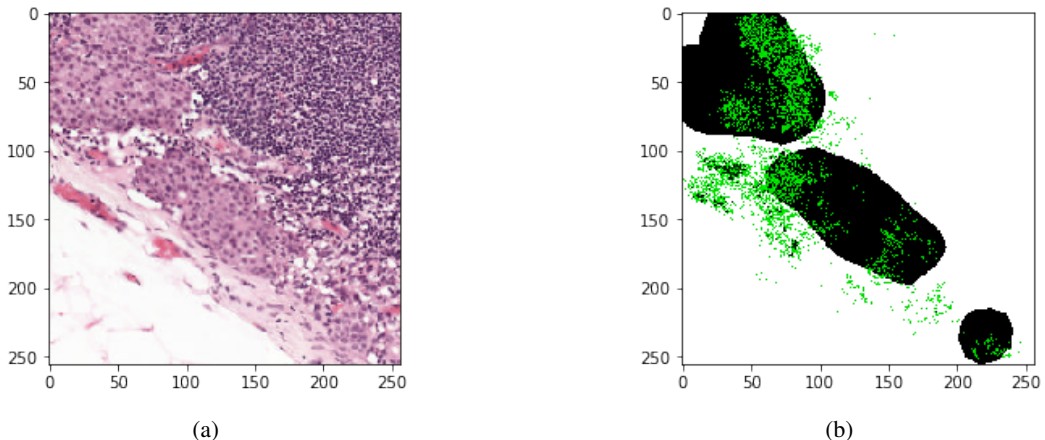

Figure 20: Example False Negative Image of nCAMELYON validation sets and corresponding segmentation map with saliency overlaid.

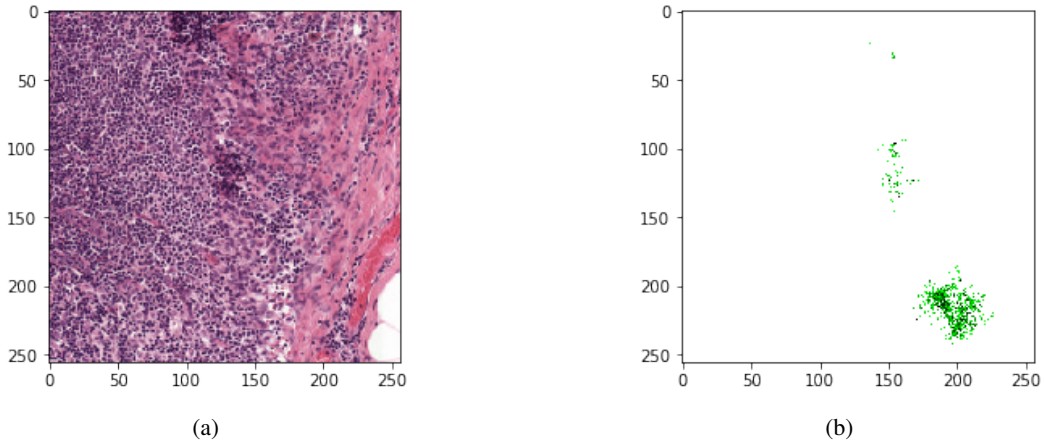

Figure 21: Example False Positive Image of nCAMELYON validation sets and corresponding saliency map.

Table 3: Schematic of the architecture of the different topological embedding encoders used in this paper. The operations and their corresponding parameters of the residual blocks are denoted in brackets. The first block within each section performs downsampling using convolutions with stride 2. We use InstanceNorm instead of BatchNorm and test different pooling methods after the topological embeddings.

| RF=9 | RF=33 | RF=177 |
|:---:|:---:|:---:|
| conv, $1 \times 1, 64$ | | |
| conv, $3 \times 3, 64$ | | |
| conv, $1 \times 1, 64$
conv, $3 \times 3, 64$
conv, $1 \times 1, 256$ | conv, $1 \times 1, 64$
conv, $3 \times 3, 64$
conv, $1 \times 1, 256$ | conv, $1 \times 1, 64$
conv, $3 \times 3, 64$
conv, $1 \times 1, 256$ |
| conv, $1 \times 1, 64$
conv, $1 \times 1, 160$
conv, $1 \times 1, 256$ | conv, $1 \times 1, 64$
conv, $1 \times 1, 160$
conv, $1 \times 1, 256$ | conv, $1 \times 1, 64$
conv, $3 \times 3, 64$
conv, $1 \times 1, 256$ |
| conv, $1 \times 1, 64$
conv, $1 \times 1, 160$
conv, $1 \times 1, 256$ | conv, $1 \times 1, 64$
conv, $1 \times 1, 160$
conv, $1 \times 1, 256$ | conv, $1 \times 1, 64$
conv, $3 \times 3, 64$
conv, $1 \times 1, 256$ |
| conv, $1 \times 1, 128$
conv, $3 \times 3, 128$
conv, $1 \times 1, 512$ | conv, $1 \times 1, 128$
conv, $3 \times 3, 128$
conv, $1 \times 1, 512$ | conv, $1 \times 1, 128$
conv, $3 \times 3, 128$
conv, $1 \times 1, 512$ |
| conv, $1 \times 1, 128$
conv, $1 \times 1, 320$
conv, $1 \times 1, 512$ | conv, $1 \times 1, 128$
conv, $1 \times 1, 320$
conv, $1 \times 1, 512$ | conv, $1 \times 1, 128$
conv, $3 \times 3, 128$
conv, $1 \times 1, 512$ |
| conv, $1 \times 1, 128$
conv, $1 \times 1, 320$
conv, $1 \times 1, 512$ | conv, $1 \times 1, 128$
conv, $1 \times 1, 320$
conv, $1 \times 1, 512$ | conv, $1 \times 1, 128$
conv, $3 \times 3, 128$
conv, $1 \times 1, 512$ |
| conv, $1 \times 1, 128$
conv, $1 \times 1, 320$
conv, $1 \times 1, 512$ | conv, $1 \times 1, 128$
conv, $1 \times 1, 320$
conv, $1 \times 1, 512$ | conv, $1 \times 1, 128$
conv, $3 \times 3, 128$
conv, $1 \times 1, 512$ |
| conv, $1 \times 1, 256$
conv, $1 \times 1, 640$
conv, $1 \times 1, 1024$ | conv, $1 \times 1, 256$
conv, $3 \times 3, 256$
conv, $1 \times 1, 1024$ | conv, $1 \times 1, 256$
conv, $3 \times 3, 256$
conv, $1 \times 1, 1024$ |
| conv, $1 \times 1, 256$
conv, $1 \times 1, 640$
conv, $1 \times 1, 1024$ | conv, $1 \times 1, 256$
conv, $1 \times 1, 640$
conv, $1 \times 1, 1024$ | conv, $1 \times 1, 256$
conv, $3 \times 3, 256$
conv, $1 \times 1, 1024$ |
| conv, $1 \times 1, 256$
conv, $1 \times 1, 640$
conv, $1 \times 1, 1024$ | conv, $1 \times 1, 256$
conv, $1 \times 1, 640$
conv, $1 \times 1, 1024$ | conv, $1 \times 1, 256$
conv, $3 \times 3, 256$
conv, $1 \times 1, 1024$ |
| conv, $1 \times 1, 256$
conv, $1 \times 1, 640$
conv, $1 \times 1, 1024$ | conv, $1 \times 1, 256$
conv, $1 \times 1, 640$
conv, $1 \times 1, 1024$ | conv, $1 \times 1, 256$
conv, $3 \times 3, 256$
conv, $1 \times 1, 1024$ |
| conv, $1 \times 1, 256$
conv, $1 \times 1, 640$
conv, $1 \times 1, 1024$ | conv, $1 \times 1, 256$
conv, $1 \times 1, 640$
conv, $1 \times 1, 1024$ | conv, $1 \times 1, 256$
conv, $3 \times 3, 256$
conv, $1 \times 1, 1024$ |
| conv, $1 \times 1, 256$
conv, $1 \times 1, 640$
conv, $1 \times 1, 1024$ | conv, $1 \times 1, 256$
conv, $1 \times 1, 640$
conv, $1 \times 1, 1024$ | conv, $1 \times 1, 256$
conv, $3 \times 3, 256$
conv, $1 \times 1, 1024$ |
| conv, $1 \times 1, 512$
conv, $1 \times 1, 1280$
conv, $1 \times 1, 2048$ | conv, $1 \times 1, 512$
conv, $3 \times 3, 512$
conv, $1 \times 1, 2048$ | conv, $1 \times 1, 512$
conv, $3 \times 3, 512$
conv, $1 \times 1, 2048$ |
| conv, $1 \times 1, 512$
conv, $1 \times 1, 1280$
conv, $1 \times 1, 2048$ | conv, $1 \times 1, 512$
conv, $1 \times 1, 1280$
conv, $1 \times 1, 2048$ | conv, $1 \times 1, 512$
conv, $3 \times 3, 512$
conv, $1 \times 1, 2048$ |
| conv, $1 \times 1, 512$
conv, $1 \times 1, 1280$
conv, $1 \times 1, 2048$ | conv, $1 \times 1, 512$
conv, $1 \times 1, 1280$
conv, $1 \times 1, 2048$ | conv, $1 \times 1, 512$
conv, $3 \times 3, 512$
conv, $1 \times 1, 2048$ |
| conv, $1 \times 1, 512$
conv, $1 \times 1, 1280$
conv, $1 \times 1, 2048$ | conv, $1 \times 1, 512$
conv, $1 \times 1, 1280$
conv, $1 \times 1, 2048$ | conv, $1 \times 1, 512$
conv, $3 \times 3, 512$
conv, $1 \times 1, 2048$ |

