# OpenReview forum: "Needles in Haystacks: On Classifying Tiny Objects in Large Images"
_ICLR.cc/2020/Conference — Reject_

### Official Review · AnonReviewer2 · 2019-10-23
**Official Blind Review #2**

**Rating:** 3

**Review:**

The submission proposes an analysis of the impact of object size in images when performing classification tasks using neural networks of the BagNet family. The analysis is performed on two datasets, a large resolution cluttered MNIST and a histopathology dataset named nCAMELYON.

The paper attack interesting questions and links the size of the object in the image (O2I) to the training dataset size required. Also, showing that max-pooling is the only pooling operation that converges for very low O2I (but is the slowest to converge at higher O2I) is interesting and encourages discussion about the training (optimization) process.

In my opinion, the main issue about the submission is the limited depth in the contributions, analyzing a single family of network architectures (BagNets) over two datasets, one of which is relatively small. The family of R-CNN and its derivatives were especially designed to counter the impact of object size, it would have been interesting to include them in the analysis. Furthermore, limited insights can be carried out for tasks related to classification such as localization and segmentation. Models such as Single Shot Detectors split the image into grids and variable anchor sizes to perform their inference, are they affected to a lesser extent by object size? The limited insights provided to potential future readers prevents me from recommending the submission for acceptance.

On p. 7 (and fig. 7 (b) ), it is said that the lower performance of the larger receptive field suggests that class-relevant information is contained in the texture. I am not sure about this remark, I would have expected the network to learn to focus on the right regions, provided the receptive field is big enough to see the whole object of interest. Could the decrease in performance be attributed to the increased amount of learnable parameters that ended up too large for the “relatively small nCAMELYON dataset used for training”? (sec. 3.1, Global pooling operations)

Fig. 16 seems to suggest that multiple numbers can overlap significantly in nMNIST, yielding potentially confusing images even for humans. I am not sure if this is a desirable characteristic for such dataset.


Minor details
- Please use \cdot instead of the asterisk operator to denote multiplication (sec. 3, footnote 3, fig. 5 (b-c) and sec. 3.1;
- Fig. 4 uses two different (and non-linear) values for the x-axes, giving the impression that both curves can be compared, that might be confusing to the reader;
- Sec 1 and 5 “pixel-level level annotations”: duplicate “level”;
- p. 6 “1 and , 2”: extraneous comma.


**Experience Assessment:**

I have published one or two papers in this area.

**Review Assessment: Checking Correctness Of Derivations And Theory:**

I assessed the sensibility of the derivations and theory.

**Review Assessment: Checking Correctness Of Experiments:**

I assessed the sensibility of the experiments.

**Review Assessment: Thoroughness In Paper Reading:**

I read the paper thoroughly.

---

> ### Author Response · Authors · 2019-11-14
> **Rebuttal to review 2**
>
> Many thanks for dedicating time to review our paper. We feel the reviewer is missing some important points of our study, which we would like to clarify below:
>
> [Depth of the contribution]
> We argue that the depth of our contribution is significant: the paper makes an important contribution by revealing a substantial limitation of current methods, and provides methodology (a testbed) to empirically study this limitation on two datasets. We would strongly argue that these contributions can help to advance the field, and should be considered as important as proposing a new method or theory. Moreover, the breadth of our analysis is substantial as for each dataset, we explored 48 different variants of ResNets (3 receptive field sizes, 4 different pooling operations and 4 different model capacities - please refer to section B in the supplementary material for the details about the tested model architectures). To ensure the robustness of our conclusions we ran more than 750 different hyperparamenter setups (each one run with 6 different random seeds). This is not a trivial empirical study, and the insights we can gain from it should be of interest to many in the community.
>
> [Suitability of object detection architectures e. g. R-CNN]
> R-CNN or SSD like architectures were built to tackle object detection problems, whereas we are interested in image classification problem. In object detection, one would require access to localized labels such as pixel level annotations in the form of bounding box information. Such annotations are not available in many important real-world applications, and hence R-CNN type methods are not applicable. Moreover, if bounding box annotations were available one could easily translate the low signal-to-noise classification problem into an imbalanced classification problem (with small number of positive classes and large number of negative classes) — as it is done in object detection approaches. Therefore, we argue that the problem we are studying in this paper is very different, but equally relevant, from the object detection setup and has different challenges which make object detection pipelines not applicable to our setup.
>
> [fig. 7 (b)]
> When designing our experiments we ensured fixed model capacity by increasing the number of filters for the networks with lower receptive fields (see Table 3 in the appendix). Therefore, all models have similar capacity (unless specifically stated). We agree with the reviewer, that when having access to large dataset one should observe good results for all receptive field sizes that are bigger than the size of an object of interest (as confirmed by our results on nMNIST). However, when dataset is relatively small (such as nCAMELYON), we observe that the small receptive field tends to generalize better. We attribute this to the fact that within a larger receptive field there are more spurious correlations (more random patterns might look like a class-discriminative ones), as a result the model with larger receptive field overfits. We clarified this in the manuscript.
>
> [Digits overlap]
> Although restricting the digits’ overlap might clarify the classification problem for humans, we argue that it is not affecting the generalization of CNNs in large O2I scenarios and as a result the digits’ overlap is not affecting our conclusions. For example, we can observe that the tested models for an O2I ratios of 20% reach very good results (90%+ of test set accuracy) - suggesting that the digits’ overlap does not impede good model generalization. Moreover, our qualitative results (shown in supplementary material) show that CNNs correctly classify images with overlapping digits (see Figures 16a, 16d, 16e, 16h, 16i, 16l, 16n) while making mistakes in images without any overlap (see Figure 16c). Thus, the change of O2I seem to affect the generalization way stronger than the digits’ overlap.
>
> [Minor details]
> Thank you for spotting the typos. We fixed those in the manuscript.

---

### Official Review · AnonReviewer1 · 2019-10-23
**Official Blind Review #1**

**Rating:** 3

**Review:**

This paper presents a testbed framework to investigate the limitations of CNN at the classification of tiny objects and the effects of signal to noise ratio has in the task. The implemented framework will be made available online upon acceptance.

I believe that the question the authors try to answer is very interesting and worth of exploration. However, I am a bit less excited about the achieved results since I consider them not to be sufficient to drive conclusions. The experiments proposed by the authors are run on two different datasets created ad-hoc: the nMNIST and the nCAMELYON, both modifications of MNIST and CAMELYON datasets. Over all the experiments run, the behaviors observed on the two datasets are not the same. The explanation provided by the authors is that since nCAMELYON is very small, results are different from the ones of nMNIST. While I consider this a valid explanation, this still limits the overall conclusions of this paper. Therefore, my main recommendation to the authors would be to identify other datasets for their experiment.

In the dicussion section, the authors argue that MS COCO is not a good candidate since the background usually contains information that allows to infer the label. I wonder if it would be possible to generate a new dataset, starting from MS COCO, that avoids this problem. Using the example of the paper, I would argue that it is possible to obtain images of outdoors with people where there are no balls. Having such a dataset would make your paper much stronger.

Finally, to some extent, I consider that your problem is strongly linked to anomaly/detection detection. In the end, this is what a needle in a haystack is. How do you position yourselves with respect to the state of the art on this topic?

**Experience Assessment:**

I have published one or two papers in this area.

**Review Assessment: Checking Correctness Of Derivations And Theory:**

N/A

**Review Assessment: Checking Correctness Of Experiments:**

I carefully checked the experiments.

**Review Assessment: Thoroughness In Paper Reading:**

I read the paper thoroughly.

---

> ### Author Response · Authors · 2019-11-14
> **Rebuttal to review 1**
>
> Many thanks for the insightful comments and highlighting the interest in the problem that we aim to address. Below we address the concerns of the reviewer:
>
> [inconclusive observations on two used datasets]
> This is indeed an important observation that motivates the need for low and very low signal-to-noise classification scenarios testbed and indicates that further research is required. We do not see this as a limitation of our paper, but rather an important aspect of the study. In this paper, we designed a testbed by including two dataset to showcase the limitation of current CNNs when working with low O2I ratios: 1) nMNIST dataset is an elementary, large scale dataset that highlights that even in relatively simple scenarios current CNNs fail to generalize well when faced with low signal-to-noise images; 2) nCAMELYON highlights how current methods behave when faced with probably the largest (although still small in machine learning terms), publicly available needle-in-a-haystack dataset. Finally, we would like to point out, that by opensourcing our testbed framework (that includes data preparation, model architectures and training scripts) we invite the community to contribute new challenging low and very low signal-to-noise classification datasets.
>
> [additional datasets]
> As mentioned in the discussion section, we initially considered other datasets to be a part of our study (including MS-COCO). However, what the reviewer suggests is not straightforward at all, since the dataset does not contain an outdoor class it wouldn't be possible to automatically get a subset of images that contain people in outdoor without the ball (it would require additional human labeling). One possibility to overcome this issue might be to use another class label as an indication of outdoor class, for example, one could create the dataset for ball vs. no ball classification using a context of people and baseball bat. However, the size of such dataset would still be small (similarly to CAMELYON dataset). Nevertheless, we adapted the discussion section to suggest such an experiment as a future work.
>
> [connections to outlier detection]
> The low and very low signal-to-noise classification setup could potentially be framed as anomaly detection, e. g. one could use images without the object of interest to fit “normality” model and, then, use this model to detect regions with anomalies in an image. In this setup, one would expect that the anomaly would coincide with the object of interest (needle in a haystack). However, if approaching this problem in an “unsupervised” way there is no guarantee that the anomaly would coincide with the object of interest. One way to overcome this issue is by incorporating the information about image-level labels (e. g. by merging outlier detection with classification). We would argue that when having access to image labels it is more natural to deal with this problem as low and very low signal-to-noise classification. Nevertheless, the idea of using anomaly based model inductive biases in CNNs is interesting and we leave it for future work.

---

### Official Review · AnonReviewer3 · 2019-10-29
**Official Blind Review #3**

**Rating:** 1

**Review:**

The authors present an empirical study to evaluate the performance of CNN-based object classifiers for situations in which the object of interest is very small relative to the size of the image. Two artificial datasets, based on MNIST and histopathological images are introduced to conduct the experiments. Through empirical evaluation the authors conclude that the size of the dataset required for generalization increases rapidly with the inverse of the O2I ratio, that higher capacity models generalize better, and that accounting for the model's receptive field is key.

The contributions of the study are limited: i) The artificial datasets generate images that are small and O2Is that are big for the applications of interest, e.g., gigapixels images in digital pathology (see Figure 1). ii) The dataset based on MNIST is perhaps too artificial (too structured), once one compares their results relative to nCAMELYON. iii) The authors only consider ResNet-50. iv) The authors do not consider multi-instance learning pooling functions, e.g., noisy or, noisy and or attention. v) The authors do not consider performance as a function of the positive instances in the image (number occurrences of 3 in the proposed nMNIST). vi) There is no methodological contribution.

**Experience Assessment:**

I have read many papers in this area.

**Review Assessment: Checking Correctness Of Derivations And Theory:**

N/A

**Review Assessment: Checking Correctness Of Experiments:**

I carefully checked the experiments.

**Review Assessment: Thoroughness In Paper Reading:**

I read the paper thoroughly.

---

> ### Author Response · Authors · 2019-11-14
> **Rebuttal to review 3**
>
> Thanks for taking the time to review our paper. We respectfully disagree with the reviewer's assessment of the paper since it doesn't do justice to paper's contributions. In the rebuttal, we show that not all mentioned limitations are valid concerns. However, before addressing the individual concerns we would like to revisit the main points of the paper and highlight our contributions: (1) We identified an unexplored machine learning problem of image classification in low and very low signal-to-noise ratios; this problem has not been discussed in the past and the community seems unaware of it; we feel this as an important contribution as proposing a new method or theory; (2) We carefully designed two image datasets with controlled signal-to-noise ratios and highlighted that CNNs struggle to show good generalization for low and very low signal-to-noise ratios even for a relatively elementary MNIST-based dataset; (3) we ran an extensive series of controlled experiments that explore both a variety of CNNs' architectural choices and the importance of training data scale for the low and very low signal-to-noise classification. To the best of our knowledge this is the first, large scale empirical demonstration of optimization and generalization problems of CNNs with very low signal-to-noise classification scenarios. Moreover, by opensourcing our testbed framework, we invite the community to contribute to low and very low signal-to-noise classification scenarios either by incorporating new models or by proposing new datasets.
>
> Below, we address the reviewer’s concerns.
> i) We use Figure 1 as a motivational example to illustrate the need for an in depth study of low signal-to-noise classification scenarios. We run our study on the range of O2I ratios that varies from 0.075% (7e-4) and 19.14% (2e-1), thus, we cover the whole range of MiniMIAS dataset and a significant portion of CAMELYON17 dataset. Since we already observed interesting trends as well as optimization and generalization difficulties in the lower spectrum of the tested range, we did not include the O2I range of [10e-6, 7e-4] as it will show similar results to 0.075%. Moreover, the scope of the study is to explore the optimization and the generalization properties of CNN as a function of signal-to-noise ratio and not image resolution. Nevertheless, our testbed can easily be extended to more challenging O2I ratios and image resolutions.
>
> ii) We would argue that there is a value in showing current methods’ limitations in a simple, large scale dataset such as nMNIST. In a series of controlled experiments, we show that even such elementary dataset is very challenging for current CNN architectures.
>
> iii) We explore 48 different variants of ResNets (3 receptive field sizes, 4 different pooling operations and 4 different model capacities - please refer to section B in the supplementary material for the details about the tested model architectures). We decided to use ResNet-like backbones due to its versatility in a variety of computer vision tasks. These networks have shown very good results in tasks such as image classification and were proven to be very effective backbones in tasks such as object detection and image segmentation.
>
> iv) We refer the reviewer to our experiments using soft attention based pooling that is based on multi-instance learning formulation from Ilse et al. 2018. For details about pooling operations considered in our study, please see section B1 of the supplementary material.
>
> v) While the question of detection multiple ‘3’s in a single image is an interesting one, the nMNIST dataset was designed to highlight the limitations of current CNN in a clean and straightforward way— we wanted to showcase an elementary scenario in which current CNN encounter generalization problems when faced with low and very low signal-to-noise scenarios.
>
> vi) We respectfully disagree. The paper makes an important contribution by revealing a substantial limitation of current methods, and provides methodology (a testbed) to empirically study this limitation.

---

### Decision · Program_Chairs · 2019-12-19

**Decision:**

Reject

**Comment:**

This paper proposes performs an empirical study to evaluate CNN-based object classifier for the case where the object of interest is very small relative to the size of the image. Two synthetic databases are used to conduct the experiments, through which the authors made a number of observations and conclusions. The reviewers concern that the databases used are too structured or artificial, and one of the two databases is very small as well. On top of that, only one network architecture is used for evaluation. Furthermore, the conclusion from two databases seem inconsistent as well. The authors provided detailed responses to the reviewers' comments but were not able to change the overall rating of the paper. Given these concerns, as well as no methodological contribution, there are general concerns from all reviewers that the contributions of this work is not sufficient for ICLR. The ACs concur the concerns and the paper can not be accepted at its current state.